# Blockchain Technology in the Food Industry: A Review of Potentials, Challenges and Future Research Directions

**Abderahman Rejeb** [1], **John G. Keogh** [2], **Suhaiza Zailani** [3], **Horst Treiblmaier** [4,*] **and Karim Rejeb** [5]

1  Doctoral School of Regional Sciences and Business Administration, Széchenyi István University, 9026 Györ, Hungary; abderrahmen.rejeb@gmail.com
2  Henley Business School, University of Reading, Greenlands, Henley-on-Thames RG93AU, UK; john@shantalla.org
3  Department of Operations Management and Information System, Faculty of Business and Accountancy, University Malaya, Kuala Lumpur 50203, Malaysia; shmz@um.edu.my
4  Department of International Management, Modul University Vienna, 1190 Vienna, Austria
5  Higher Institute of Computer Science El Manar, 2, Rue Abou Raïhan El Bayrouni, 2080 Ariana, Tunisia; karim.rejeb@etudiant-isi.utm.tn
*  Correspondence: horst.treiblmaier@modul.ac.at

**Abstract:** Blockchain technology has emerged as a promising technology with far-reaching implications for the food industry. The combination of immutability, enhanced visibility, transparency and data integrity provides numerous benefits that improve trust in extended food supply chains (FSCs). Blockchain can enhance traceability, enable more efficient recall and aids in risk reduction of counterfeits and other forms of illicit trade. Moreover, blockchain can enhance the integrity of credence claims such as sustainably sourced, organic or faith-based claims such as kosher or halal by integrating the authoritative source of the claim (e.g., the certification body or certification owner) into the blockchain to verify the claim integrity and reassure business customers and end consumers. Despite the promises and market hype, a comprehensive overview of the potential benefits and challenges of blockchain in FSCs is still missing. To bridge this knowledge gap, we present the findings from a systematic review and bibliometric analysis of sixty-one (61) journal articles and synthesize existing research. The main benefits of blockchain technology in FCSs are improved food traceability, enhanced collaboration, operational efficiencies and streamlined food trading processes. Potential challenges include technical, organizational and regulatory issues. We discuss the theoretical and practical implications of our research and present several ideas for future research.

**Keywords:** blockchain technology; food supply chain; potentials; challenges; systematic literature review; bibliometric analysis

## 1. Introduction

The globalization of food supply chains (FSCs) and markets has led to a significant increase in products and information movements between countries [1]. Traditional FSCs are characterized by strong vertical integration and coordination among supply chain partners to promote efficiency, for example, by lowering transaction, operating and marketing costs and fulfilling consumer needs for food quality and safety [2]. Therefore, FSC exchange partners have found themselves under increasing pressure to improve the transparency of their supply chains, enhance the exchange of trusted

information, and improve the tracking and tracing capability (henceforth traceability) of agricultural products from farms through to retailers [3–5].

Additionally, the traceability of food products and overall supply chain transparency has become critical due to multiple scandals occurring in global FSCs (e.g., the horsemeat scandal in Europe, the melamine scandal in China). The need for effective traceability has intensified as regulations require that every ingredient of a food product is traceable to its source [6]. Consumer demand has led to the year-round availability of many agricultural products and intensified the pressure on businesses to provide details on product-specific attributes such as quality, safety, authenticity, traceability, provenance (food provenance is the geographic source or origin as determined by analytical science, and differs from data provenance) and conditions of production and supply [7,8]. The heightened demand for information is a driving factor for the introduction of new technologies. For example, radio-frequency identification (RFID) technology has been deployed in FSCs to aid visibility and traceability [9], reduce food waste [10], facilitate forward tracking [11–13], increase operational efficiencies [14–16], automate data collection, prevent errors in order picking and shipping [17] and intelligently control conditions (e.g., temperature, humidity) and supply chain processes [18–20]. Cloud computing platforms are used for storing information related to food products, and this information is made accessible to retailers and consumers through websites or barcode scans using a mobile device [21]. Srivastava and Wood [22] note that cloud computing enables short messaging services in agricultural supply chains, providing information about weather conditions, proper use of pesticides, alerts of disease outbreaks and government subsidies. While these platforms drive FSCs toward a digital and data-driven food ecosystem, several fundamental problems remain unaddressed. For instance, there is a lack of continuous monitoring of the FSC and an inability to predict the remaining shelf life of fresh produce [23]. Similarly, the conventional food supervision system suffers from data fragmentation, a lack of transparency caused by data discrepancies and inconsistencies, insufficient interoperability and lack of information traceability [24]. To address these problems, FSC scholars and practitioners envision the application of blockchain technology in the food industry to revolutionize the way FSCs are designed, developed, organized, and managed. According to Wang et al. [25], blockchain can potentially impact future supply chain practices and policies by providing extended visibility and traceability. Likewise, it has the potential to improve traditional supply chain processes that are characterized by a dominating actor serving as a central third-party provider imposing their own rules, governance mechanism and centralized architectures [26].

Blockchain is defined as "a digital, decentralized and distributed ledger in which transactions are logged and added in chronological order with the goal of creating permanent and tamperproof records" [27] (p. 574). Rejeb et al. [28] argue that blockchain is a combination of multiple technologies, tools, and methods to address a particular problem or business case. Aside from being a driving force behind cryptocurrencies, blockchain has gained widespread popularity in the supply chain and logistics community because of its ability to increase transparency, ensure the immutability of transactions and enhance trust among participating food stakeholders [29,30]. Since research on the integration of blockchain technology into the FSC has only recently started to emerge, there is a considerable demand for investigating its potentials for FSCs. The intricate complexity of FSCs brings about new health and safety challenges to which food stakeholders need to react by ensuring sustainable food ecosystems. For example, Dubai uses blockchain and other Internet-based technology to enhance food safety and provide consumers' with nutritional information through its "Food Watch" initiative—a technology platform that digitizes information and digitalizes food safety processes and roles as well as providing nutritional information of all edible items served through the 20,000 or more food establishments [31].

Additionally, Walmart, IBM, and Tsinghua University explored the use of blockchain to improve food safety across China and enhance the traceability of food items along the supply chain [32]. Similarly, Chinese retailer Jindong has partnered with Kerchin, an Inner Mongolia-based beef producer, to apply blockchain technology in compiling digital product information such as farm details, batch numbers,

factory and processing data, expiration dates, storage temperatures and shipping details that are digitally connected to trace every step of in the processing of the food items. Their system enables customers to trace information about frozen meat, such as a cow's breed, weight and diet, and the location of farms by scanning the QR code available on the packaging [33]. On a larger scale, according to Edwards [34], Alibaba has launched an initiative to collaborate with Blackmores and several other Australian and New Zealand-based food producers and suppliers to prevent the rise of counterfeit food items sold across China through the application of blockchain. Slovenia-based Origin Trail, a not-for-profit technology developer, created an open-source data protocol (or middleware) based on the GS1 supply chain standards that act as a standards-based interoperability platform between blockchains and legacy systems [35]. Origin Trail has partnered with the British Standards Institute (BSI) to advance blockchain use cases, especially in the food industry [36]. On a global scale, and due primarily to the multitude of use cases, no comprehensive roadmap exists to streamline blockchain implementations and adoption of blockchain-based platforms across FSCs is lagging expectations [37].

To shed light on the potentials and challenges of blockchain in the FSC, in this study, we review the state-of-the-art of the technology, its recent developments, and the applications in the food industry. Moving beyond the discussion of whether FSC stakeholders should adopt blockchain technology, we investigate the opportunities resulting from blockchain technology applications already adopted in FSCs. More specifically, we seek answers to the following research question: *What are the potentials and challenges of blockchain adoption in the FSC?*

More specifically, the literature review presented in this paper

(1) provides a background of blockchain technology to allow researchers from different fields to position their research activities appropriately,
(2) summarizes existing research and developments concerning the implementation of blockchain technology toward sustainable FSCs by outlining the potentials and challenges and
(3) identifies gaps in current research that highlights areas for further investigation.

The remainder of this paper is structured as follows. Section 2 presents the methods applied in this research for literature collection and selection in Section 3. Section 4 provides a detailed discussion of the findings of this review, and in Section 5, we answer the research questions of this study. Finally, we conclude the paper, highlight the theoretical and managerial contributions and outline the study limitations and future research directions.

## 2. Research Methodology

We conducted a systematic literature review (SLR) to identify, evaluate and interpret research and developments relevant to the application of blockchain technology in the FSC. An SLR is a rigorous and replicable method [38] to assess and analyze previously published work relevant to a particular research question, research topic or other matter of interest [39]. The systematic process of literature collection and analysis is useful for extracting pertinent insights based on the findings of previous research and identifying possible knowledge gaps [40]. We precisely followed the guidelines of Tranfied et al. [40] and Aguinis et al. [41] for traditional qualitative reviews and supplemented our review with a co-citation network analysis. This method has three stages, namely (1) planning the review, (2) conducting the review and (3) reporting the review. The following subsections elaborate on each of these stages.

### 2.1. Planning the Review

Two methodical approaches were employed to answer our research question. At the initial stage, and in line with the goals of this project, we decided to employ a qualitative method to obtain a deeper understanding of the core issues regarding blockchain technology and FSC research. Then, we decided to carefully examine several knowledge domains of blockchain and FSC research by conducting a co-citation network analysis to reveal different domains, pending issues and future research directions.

## 2.2. Conducting the Review

Scopus was used to obtain all pertinent articles from various disciplines or fields studying blockchain technology in the context of the FSC. Scopus is more comprehensive than the Web of Science (WoS), containing 84% of the WoS titles [42] and offering greater coverage of open access journals [43] including those indexed in DOAJ and other leading databases, such as IEEE Explorer, Springer, ScienceDirect and Taylor and Francis. According to Tober [44], Scopus is considered the most powerful search engine to get an overview of a particular topic. In terms of volume, Scopus contains more than 20,000 peer-reviewed journals from 5000 publishers and 1200 open access journals [45]. To collect and extract relevant articles, we initially created a pool of keywords and agreed-upon search criteria. First, the term "blockchain" was used in combination with terms that represent the FSC, including; *"food" OR "agriculture" OR "agri-food" OR "agro-food" OR "farming" OR "cold chain*" OR "fresh product*" OR "agri-fresh" OR "vegetables" OR "fruit*" OR "perishable."* The keywords were searched for in "article title, abstract and keywords." For transparency and clarity, the advanced search function used in Scopus is shown in Appendix A.

The scope of data collection was identified using several attributes, such as discipline, language, the period of publication, and document type [41]. In terms of disciplines, we restricted our search to business, computer science, engineering, decision sciences, social sciences, agriculture, environment, and economics. We only selected articles written in English and published in peer-reviewed journals. In doing so, we ensured that the reviewed literature originated from rigorous academic sources [46] and maintained a high quality of the retrieved publications. The publications were then scrutinized and treated independently and coded as (1) relevant, (2) irrelevant or (3) doubtful. After further screening, sixty-one (61) full-length articles were confirmed as the final dataset in the research.

## 3. Descriptive Results and Knowledge Domains

### 3.1. Publications by Year

The publication dates of the sixty-one (61) articles confirm the growing interest in this research area. Although blockchain technology emerged in 2008 as the underlying operating platform for Bitcoin [47], academic literature related to non-financial applications of blockchain has appeared only in more recent years. Recently, blockchain technology has been widely applied in fields such as healthcare [48–53], supply chain management [25,27,35,54–56], tourism [57–62], identity management [63–65], computer science [66], marketing [67,68] and smart cities [69]. The first publications in the food industry emerged from 2017 onwards, with most articles published in 2019. There is a sharp increase in the number of articles published between 2017 and 2019, as shown in Figure 1. The data indicate that this evolution will continue in the next few years as the technology matures and awareness of its potentials is heightened.

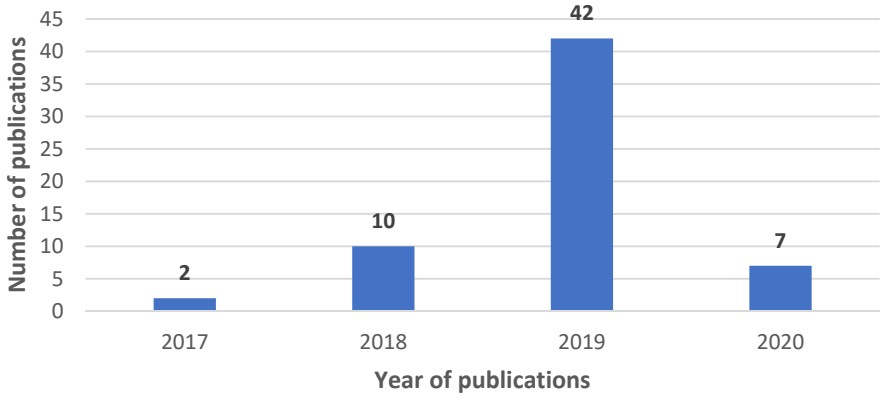

**Figure 1.** Year-wise publications.

### 3.2. Publications by Country

A significant number of the selected publications come from the USA and India, with 20 and 10 articles, respectively (see Figure 2). This is not surprising as several early blockchain adopters originate from the US, such as Walmart, who conducted trials to track pork in China and mangoes in Mexico [70]. Moreover, the USA is emerging as a leader in connecting blockchain technology with the food industry, thanks to the efforts of several companies applying blockchain to improve supply chain traceability systems. The proliferation of research and development activities can be explained by increasing food safety concerns and expectations that blockchain will promote transparency and facilitate more effective recall, resulting in greater trust in FSCs. Similarly, in India, the priority is on the agricultural sector, which accounts for nearly 18% of GDP, making the country the second-largest producer of agricultural products in the world [71]. Not surprisingly, Chinese and Italian scholars contributed substantially to the blockchain and FSC literature. In the case of China, improved food traceability is needed to enhance trust after a decade-long series of food fraud and food safety scandals. In Italy, pending problems in the agricultural FSCs such as fragmentation, lack of transparency and traceability, economic and financial waste, food fraud and food safety threats have contributed to the serious consideration of blockchain technology [72,73].

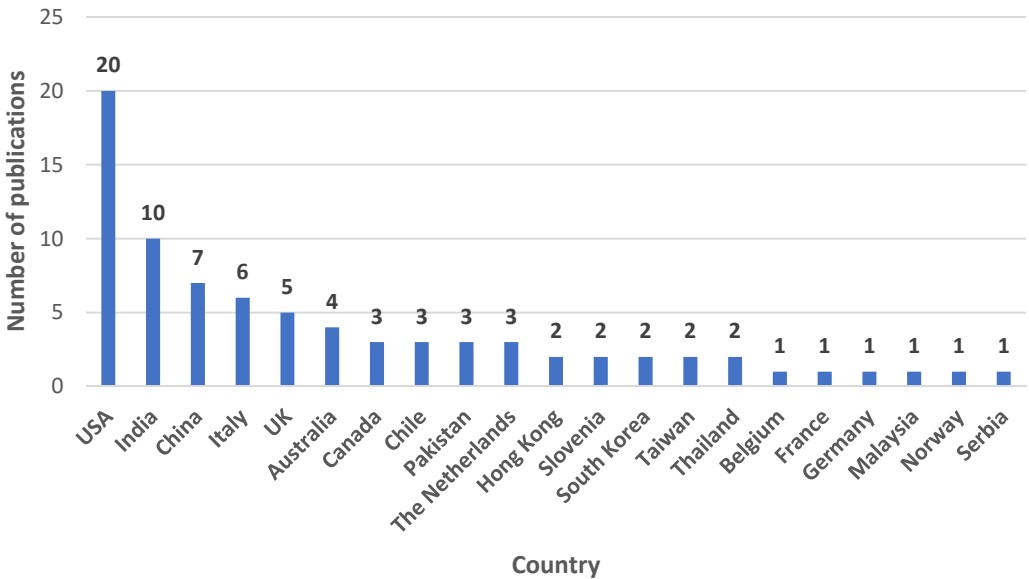

**Figure 2.** Country-wise publications.

### 3.3. Publications by Journal

Research on blockchain applications in the FSC is published in high-quality peer-reviewed journals such as IEEE Access, The International Journal of Information Management, Computers and Electrical Engineering and IEEE Internet of Things (see Figure 3). Overall, the reviewed articles were published in 48 different journals with the IEEE Access journal leading and followed by the International Journal of Advanced Computer Science and Applications and the International Journal of Information Management. Interestingly, the content of the 48 publication outlets spans across a wide variety of disciplines, including business, computer sciences, management, supply chain management, and information technology.

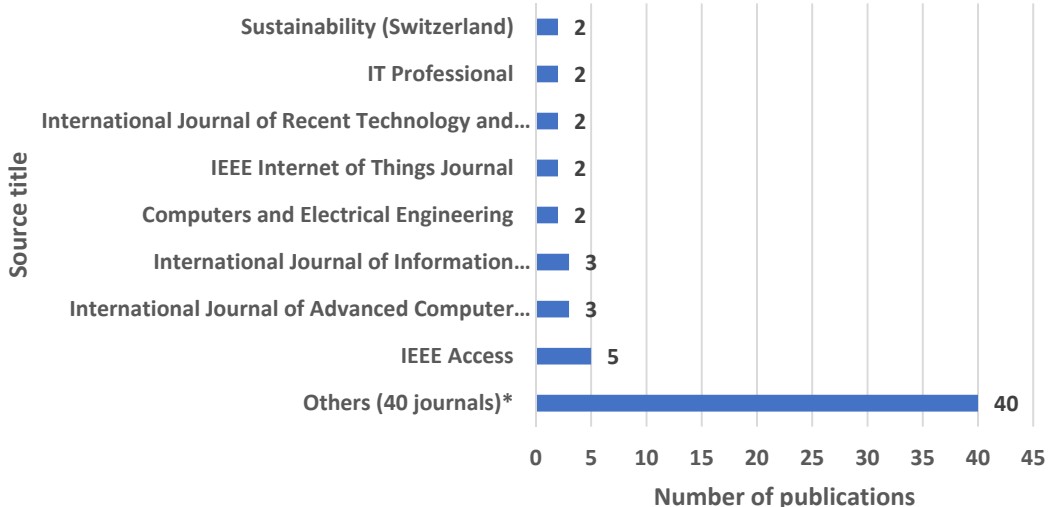

**Figure 3.** Journal-wise publications.

### 3.4. Bibliometric Analysis

The selected publications can be analyzed using several bibliometric methods and techniques such as citation analysis, co-citation analysis and co-authorship [74,75]. In this study, we review blockchain technology applications in the FSC and provide readers with an overview of current blockchain and FSC research. Using a scientometric analysis allows us to visualize knowledge in a way that is easy to interpret, highlight scholarly communities, discover knowledge domains, identify trends in different research areas and reveal relationships among scholars and institutions [76]. Several software packages are available to conduct bibliometric analyses, including BibExcel, VOSviewer, UCINET, CiteSpace and Gephi. In this study, we used VOSviewer to generate the keyword co-occurrence and bibliographic coupling networks. VOSviewer specializes in network visualization. It is a powerful tool to analyze many types of bibliometric networks, ranging from citation networks between documents and co-authorship networks between scholars to keyword co-occurrences [77]. In our study, network nodes represent either a keyword or an article. The color of the node reflects a particular property, and its radius indicates the frequency of a keyword or how often a document is cited.

#### 3.4.1. Keyword Co-Occurrence

The analysis of keyword co-occurrence helps to identify the primary topics discussed in a particular research area by visualizing similarities among frequently co-occurring keywords or topics in the literature ([78]. The co-occurrence describes the number of times two words appear together in the title, abstract or list of keywords [79]. Applying this bibliometric technique, researchers can get a broad picture regarding the content of a paper, including its methods, objectives, and viewpoints. Thus, the analysis of keyword co-occurrence is critical to examine current topics and developments associated with blockchain technology applications in the FSC. The original data were prepared, and similar keywords, such as "*blockchain*" and "*blockchain technology*," were merged to generate the keyword co-occurrence network. After fixing the threshold of keyword co-occurrence at a minimum of two, the visualization of content results in 35 nodes with different colors, as shown in Figure 4. Each node in the figure represents a keyword, and the radius of the node corresponds to its frequency in the literature. Keywords that co-occur frequently tend to be located next to each other in the network. As shown in Figure 4, the keywords were grouped into four main clusters with a different level of significance.

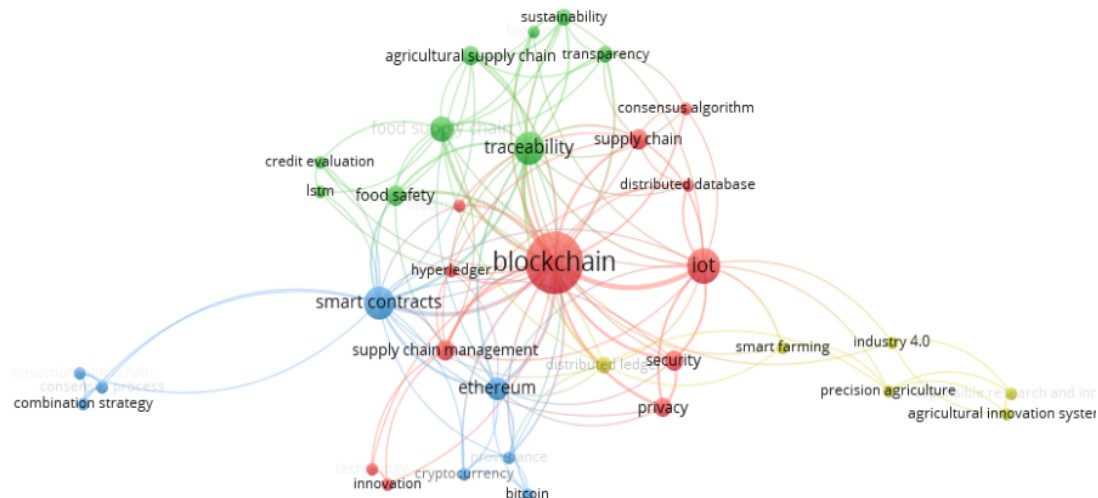

**Figure 4.** Keyword co-occurrence network.

In terms of size, the central red cluster is the most significant one, and it contains several terms strongly associated with the underlying technology blockchain. These include "*blockchain*", "*IoT*", "*supply chain*", "*consensus algorithm*", "*security*", "*privacy*", "*distributed database*" and "*hyperledger*". Most of these terms denote the technical characteristics and features of blockchain. As an example, the Internet of Things ("*IoT*") co-occurred frequently with "*blockchain*", indicating that this technology will increase the accuracy and speed of capturing information, help to manage data communication effectively and substantiate the value of blockchain technology for FSCs. The second cluster is the green one, and it consists of terms such as "*traceability*", "*food safety*", "*agricultural supply chain*", "*transparency*," and "*sustainability*". Combining these keywords reveals several critical use cases of blockchain technology in the FSC, such as traceability, which is identified as the primary driver for blockchain adoption in agricultural supply chains [80]. The blue cluster is related to applications of blockchain technology in cryptocurrencies and smart contracts. This cluster represents the first and second generations of blockchain applications for providing a ledger that records cryptographically signed transactions and offers a general-purpose programmable infrastructure using smart contracts [81]. The use of cryptocurrencies and smart contracts unlocks several potentials in the FSC because they help automate business processes and initiate transactions among FSC entities, resulting in better coordination and optimization of the entire FSC [30,82,83]. Finally, the yellow cluster includes keywords such as "*smart farming*", "*precision agriculture*", "*agricultural innovation system*," and "*industry 4.0*". Terms that appear in this cluster point toward blockchain technology to advance the digitization (i.e., from analog to digital) and business process modernization (i.e., digitalization) of the agricultural sector.

### 3.4.2. Knowledge Domains through Bibliographic Coupling

In this section, we elucidate several knowledge domains related to blockchain literature in the context of FSCs. Bibliographic coupling is conducted for all reviewed articles that cite the same literature. This implies that the higher the similarity of the referenced literature, the more similar the research content of the two articles is [84]. Knowledge mapping using bibliographic coupling analysis is an approach to study the intellectual structure of blockchain-enabled FSCs. Figure 5 presents the bibliographic coupling network of all reviewed articles, with each node representing an article. To enhance the visualization of the chart, the article titles were coded from A1 to A60. Table 1 lists the authors of the respective articles. The upper part of Figure 5 shows the co-citation clusters. In setting the threshold of bibliographic coupling to a minimum of two, four clusters were generated. The color and size of a node reflect the cluster it belongs to and its degree of centrality, representing the number of links connecting an article with related ones.

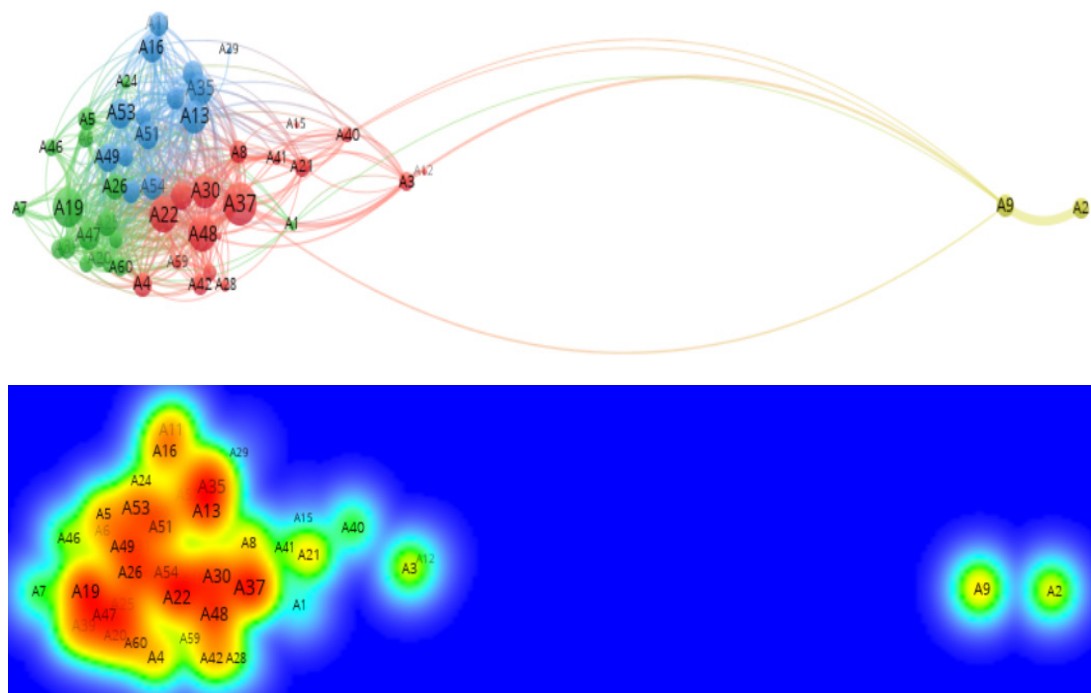

**Figure 5.** Knowledge domains in blockchain-enabled food supply chains (FSCs).

**Table 1.** Bibliographic coupling of the reviewed articles and their respective clusters.

| Cluster | References |
|---|---|
| Red | [1,80,84–100] |
| Green | [3,30,101–115] |
| Blue | [47,73,82,83,116–125] |
| Yellow | [126,127] |

The node size is proportional to the number of references a citing paper shares with other papers, thus indicating knowledge domains. A small distance between the two nodes shows that the papers are highly connected. The lower part of Figure 5 illustrates the generated clusters through a so-called "heat map." In this respect, bold fonts and warm colors (i.e., red, yellow) indicate that nodes located in the respective areas are valuable and influential. Table 1 lists the respective articles according to their cluster classification.

Based on the findings of the bibliographic coupling of the articles, four clusters emerged that were identified after reviewing the titles, abstracts, and keywords of articles in each cluster. To attribute a theme to each cluster, two authors independently reviewed the articles of each cluster and proposed an overarching topic. Typically, one cluster contains several themes, but one dominant theme often prevails and determines its overall structure. Therefore, after several rounds of discussions, the authors identified the dominant themes.

Starting at the bottom left corner of the network in Figure 5, a major part of nodes forming the knowledge domain in green represent articles related to blockchain, IoT, smart contracts and other technical characteristics. The heat map reveals that the concentration of research occurs around the nodes A19 [114], A26 [110] and A47 [3]. These articles establish the conceptual foundation to understand blockchain technology, its working mechanism and its combination with IoT. The second concentration appears around the nodes A22 [100], A30 [89], A37 [80] and A48 [1]. As indicated by the blue nodes, the third knowledge domain includes academic literature dealing with blockchain applications in the food trade, agriculture, and the sustainable operations of FSC processes. Taking into consideration the radius of nodes and their position on the heat map, nodes such as A13 [116], A16 [83],

A35 [122], A51 [82], A53 [125] are the most popular works in this knowledge domain. The central themes in this cluster are the role of blockchain in improving food trade, supporting the transition toward agriculture 4.0, and enabling the development of sustainable FSCs. Other nodes in this cluster emphasize efficiencies [117], automation [115] and food safety [123] brought by blockchain into the FSC.

The nodes scattered at the right corner of Figure 5 show the last knowledge domain, which has the least number of nodes with a low degree of centrality. Research in this cluster studies the impact of blockchain adoption on financial transactions, agricultural activities and FSC operations. However, this cluster's overall influence is considerable compared to those of the other knowledge domains, as is illustrated by the small size of the nodes. In the next section, we go into further detail and provide an in-depth discussion of the possibilities and challenges of blockchain adoption in the FSC.

## 4. Discussion

Figure 6 presents a conceptual framework that highlights the potentials and challenges of blockchain in the FSC. As for the potentials, food traceability represents the foundation of increasingly sophisticated, industrialized and globalized food value chains [128] because it helps to ensure food safety and quality, thereby fulfilling consumer expectations and demands [129]. Moreover, blockchain supports FSC collaboration and resource sharing, strengthening relationships and trust between FSC partners and may lead to quality improvements and innovation. Supply chain efficiencies are at the core of sustainable food security, and blockchain technology holds the potential to reduce transaction costs and increase overall efficiency and supply chain resilience in the food industry. Food trade is an essential economic activity that can be facilitated by the implementation of blockchain. As such, the globalization of FSCs has posed additional challenges for businesses due to the need to ensure trust, transparency and security in food trade processes. Despite several benefits of blockchain, the implementation of the technology in FSCs does not come without its drawbacks. Technical, organizational, and regulatory challenges constitute significant barriers that impede blockchain adoption and diminish its potentials for FSCs. In the next subsections, we provide a more detailed discussion of the core elements of our framework (see Figure 6).

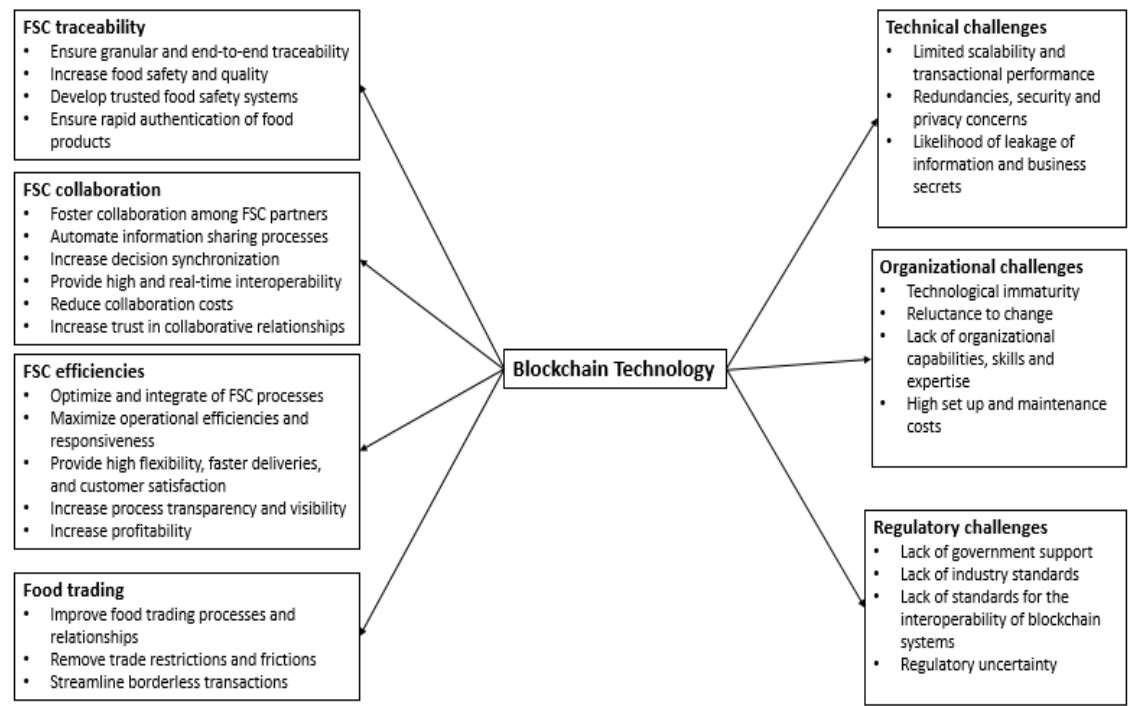

**Figure 6.** A conceptual framework for the literature analysis.

### 4.1. Potentials of Blockchain Technology in the FSC

#### 4.1.1. Food Traceability

According to Bosona and Gebresenbet [130], food traceability is part of logistics management. It deals with the capture, storage, and transmission of information about food products throughout the stages of the FSC in order to facilitate the control of food quality and safety and ensure the backward tracing and forward tracking of the food. Traceability is increasingly regarded as a vital aspect of providing safe and wholesome food [131–133] and the assurance of consumer satisfaction and trust. The need for food traceability is clearly emphasized among consumers, particularly after consecutive scandals around the world [134,135], such as the melamine crisis in China, the European horsemeat scandal [136], and the Bovine Spongiform Encephalopathy (BSC) crisis [137]. As a result, stronger regulations demand the introduction of food traceability systems and represents an opportunity for FSC exchange partners to share essential data and information about food products [130,138–144]. Food traceability benefits consumers, the food industry, retailers and regulators alike [145]. Moreover, blockchain technology can provide supply chain traceability and information transparency and enable the rapid identification of the history, movement, and current location of consumer products (or a lot or batch) with food safety issues [3]. FSC trading partners will be able to maintain a transaction recording system that can be used to aid the integrity of food products as they move along the supply chain, helping to increase control over all transactions and interactions between food suppliers, producers, logistics providers and customers [112].

Blockchain technology can add substantial value to food businesses as traceability of end products is feasible at every point in the FSC, with the ability to rapidly retrieve all data related to food in a matter of seconds [97]. For example, Lin et al. [82] propose a food safety traceability system that is based on blockchain technology and EPCIS (i.e., a GS1 global standard enabling interoperability, which is also the accepted ISO standard) to simplify the process of acquisition, management and exchange of product information. In this system, consumers can trace information about the food they purchase through the consumer traceability client application. Similarly, Hao et al. [117] propose another blockchain system for the traceability of agricultural products. Combined with IoT sensors, blockchain technology allows stakeholders to provide retrievable data storage records of agricultural products. Salah et al. [122] suggest an approach that leverages the Ethereum blockchain and smart contracts to optimize the food traceability processes of soybean across the agricultural supply chain. All events and transactions are recorded and stored in the blockchain system, providing a high level of data and information transparency and product traceability. Likewise, He et al. [118] develop a decentralized and non-reversible traceability system for optimizing commodity data storage. Alonso et al. [101] present a platform that combines IoT, edge computing, artificial intelligence and blockchain technology to manage farming environments. When using blockchain technology, all data generated by IoT sensors are securely recorded for traceability purposes. The goal of maintaining end-to-end food traceability within a very complex and fragmented FSC is no longer a challenging task using blockchain because it creates a common platform for data collection all along the FSC. Blockchain makes it possible to rapidly track (forward) and trace (backward) all batches of products and to safely withdraw from the market if unsafe or non-compliant with regulations [120]. The application of blockchain improves FSC management processes and helps food operators to differentiate products based on their quality attributes. Unlike traditional food traceability systems, blockchain technology can ensure the capture of all traceability records covering all critical information exchange points between stakeholders in the FSC [89]. Therefore, blockchain technology can increase consumer confidence in the quality, safety, authenticity, and provenance of food products and related data and information integrity.

It should be noted explicitly that food quality, safety, authenticity, and provenance must be validated through analytical science methods. For example, data provenance is often confused in blockchain marketing with the scientific provenance of food products. Scientific provenance (geographic source or

origin) can only be guaranteed through analytical science methods, such as the Carbon-13 ($^{13}$C) analysis of the food or through other scientific techniques [146]. Furthermore, documents (e.g., certificates, credence claims verification such as organic, halal, kosher) from analytical science testing can be added to a blockchain to enhance the data records' overall integrity. This is especially true if the authoritative source (e.g., a laboratory or certification body) directly adds the documents to the blockchain as they cannot be altered [147]. For FSC trading partners, access to a blockchain-enabled traceability system with enhanced data integrity for food safety, product authenticity and provenance might be highly valued and used as a driving force to boost competitiveness and increase consumer purchases [47].

### 4.1.2. FSC Collaboration

Increased supply chain collaboration and integration have led to more flexible information sharing [148], a more cohesive market focus, better coordination of sales and demand fulfilment and fewer risks related to demand uncertainty [149]. Um and Kim [150] conclude that supply chain collaboration improves firm performance and leads to transaction cost advantages. In an FSC context, supply chain collaboration is crucial because the agri-food industry structure is inherently complex. For example, multi-ingredient food products usually comprise an intricate supply chain with several business entities and multiple information exchanges. Accordingly, when multiple exchange partners form a specific FSC, interoperability issues arise due to non-compatible systems required for information sharing. To address this issue, blockchain technology platforms can utilize GS1 standards to facilitate interoperability among the different parties of the FSC. On 10 June 2020, the GS1 USA organization announced a successful proof of concept for data sharing was completed between four competing platforms and solution providers (IBM Food Trust, SAP, RIPE.IO and FoodLogiQ) [151]. According to Bumblauskas et al. [86], blockchain is a suitable solution for FSCs and agriculture because of its ability to share immutable data between exchange partners and automate information sharing processes.

The generation of rich data in the FSC is well suited for blockchain to support the collection, storage and visualization of information. Moreover, blockchain technology is expected to be a basic data-driven collaboration framework which shortens the FSC and offers new opportunities for information sharing and efficient decision-making [91]. To streamline the beef FSC processes, Surasak et al. [152] develop a system that uses blockchain to share information related to location tracking, temperature, humidity and ownership transfers. Companies advocating greater FSC transparency also envision blockchain as a potential solution for accelerating across a vast network of trusted exchange partners. For example, IBM launched its IBM Food Trust platform, which represents a cloud-based, permissioned blockchain solution for ensuring a trusted method for participants to share food-related data, extract value from others' contributions, and develop a safer, smarter, more efficient and sustainable food ecosystem [153]. Blockchain technology is a crucial enabler in the FSC, acting as the facilitator of business process and role automation (e.g., digitalization), information sharing and enhanced decision-making. The reviewed literature illustrates the virtues of blockchain technology by increasing supply chain visibility and optimizing information flows. For instance, Perboli et al. [120] argue that blockchain can guarantee end-to-end integration and transfer of information flows associated with items and batches by driving high and real-time interoperability within existing Enterprise Resource Planning (ERP) software. Similarly, Kamble et al. [91] note that blockchain can overcome several efficiency and transparency issues in the agricultural supply chain as the technology can consolidate links between producers, farmers and markets. Furthermore, FSC stakeholders can optimize several functions such as the movement of highly perishable food products through the supply chain network and the fast and targeted removal of foods unfit for consumption. More specifically, blockchain aids in establishing cooperative FSCs by lowering collaboration and administrative costs, such as costs for the design of collaborative agricultural activities, the sharing of equipment, tools and transportation [107]. The focus on developing more synchronized and collaborative relationships between FSC partners is fostered by blockchain because of its ability to secure resource exchanges and increase trust [1].

### 4.1.3. FSC Efficiencies

As competition has intensified within and between FSCs, food organizations must maintain higher levels of efficiencies and deliver added value to consumers [90]. Companies that can create more efficient FSCs will be able to achieve a significant competitive advantage. The digitization (and digitalization) of supply chains has brought tremendous opportunities for companies and created new opportunities and business models [154]. Accordingly, the evolution of FSCs has necessitated the deployment of modern technologies to enable process and role automation between information exchange partners. The adoption of blockchain technology can improve inbound efficiency and optimize planning decisions by providing reliable data and information and increasing the visibility of supply chain inventory and processes [120]. The critical assumption here is that the exchange parties do not collude to enter false data into the blockchain, which is a potential threat to the integrity of the FSC. Technology is particularly crucial for integrating FSC processes, accelerating the flow of information (i.e., transparency relates to visibility and flow of information) among FSC partners, and maximizing the efficiency, responsiveness and resilience of the food chain to account for changing market conditions and potential system shocks (e.g., COVID-19 pandemic). Food enterprises and logistics service providers can significantly reduce the inefficiencies resulting from paperwork and other fragmented and bureaucratic procedures in the supply chain.

Blockchain contributes to the automation of organizational processes and encourages firms to engage in more efficient FSC collaboration. As a result of these benefits, lower costs can be achieved through greater efficiency and better access to reliable information. Blockchain can reinforce multi-party trust and be combined with other established B2B technologies such as EDI, XML and API-based B2B [47]. The use of blockchain within these systems helps to enhance big data integration and automate certain activities within food safety management governance systems. It plays a more significant role in increasing the flexibility of the FSC, the efficient flow of information and materials through the chain [120], ensuring fresher products and faster deliveries, reduced stock levels and quick response to consumer demands and concerns. In the case of issues related to food safety, retailers can quickly and efficiently trace back along the products' supply chain to remove a specific batch of contaminated food instead of recalling the entire inventory [89]. The speed of locating food products can be done within seconds on blockchain compared to the same activity in a non-integrated system [90]. This capability is beneficial in food poisoning and food fraud, as a recall action of a suspicious item is mandatory by law and necessary to increase confidence in the brand [89]. Implementing blockchain in food recall processes in multi-party supply chains can further help save costs due to the specificity or granularity of the recorded transactions. Consequently, food companies can prevent defective or unsafe food distribution and mitigate potential economic losses and reputational damage [110]. Therefore, the integration of blockchain into the food industry gives way to a new information architecture for FSCs that will replace existing siloed databases and fosters sharing all events carried out across the supply chain, resulting in streamlined FSC processes.

### 4.1.4. Food Trading

Food is a unique commodity, and its trade has significant economic importance for both developed and developing countries [155]. For example, China's food trade significantly contributes to its national economy, the progress of agricultural innovation activities, and the population's nutritional health status [156]. Although the rapid growth of international food trade drastically changed the global food system over the past decades [157] and still exerts a considerable impact on sustainable economic development, there are multiple technological problems in today's FSCs, according to Mao et al. [83]. To address these issues and promote food trade practices among the many FSC stakeholders, blockchain technology can be used to enforce automated trading mechanisms [83].

The application of blockchain brings enhanced optimization and automation to business processes and roles between trading relationships and increases trust in exchange transactions. The auditability of blockchain technology is a crucial feature by which FSCs can establish an authoritative record for trade

data and information, perform real-time order checking and achieve visibility regarding inventory levels and automatic reconciliation of invoices [91]. Moreover, blockchain can expand food trade through the removal of certain trade restrictions. It can accelerate commercialization processes—blockchain guarantees all FSC exchange partners can attain information symmetry, credibility and trust [125]. Similarly, several companies consider blockchain a workable solution for managing trade-related documentation and streamlining borderless transactions [3]. Global FSC trading partners can benefit from blockchain's ability to reduce logistics costs, streamline transportation processes and formalize trade relationships. Also, blockchain has the potential to dismantle specific trade barriers and create new market opportunities for firms to grow, thrive and become more competitive. Mao et al. [125] note that blockchain can open up new avenues for trade that extend beyond national boundaries by enhancing traceability and efficiency in trade and addressing food safety issues. Blockchain technology represents a significant paradigm shift in terms of how transactions are conducted [3]. Therefore, the implementation of blockchain in the FSC can enable organizations to perform business beyond existing corporate boundaries almost as effectively and efficiently as they operate within the firm.

*4.2. Challenges of Blockchain Technology in FSCs*

4.2.1. Technical Challenges

According to Behnke and Janssen [1], several technical issues of blockchain remain unresolved, despite the advantages it offers in FSCs. For example, scalability needs to be addressed because the technology might become inefficient if the number of FSC transactions is increasing exponentially [114]. The validation process of transactions might limit the applicability of blockchain and reduce transaction efficiency in situations of high transaction throughput [125]. Wu et al. [114] argue that blockchains are not suitable for FSCs in general because the technology has a limited capacity to handle and store massive amounts of data. The authors further note that a multi-tier supply chain network necessitates the processing of many transactions in a short period, and blockchain technology can lead to transaction inefficiencies as well as redundancies. Perboli et al. [120] stress that the performance of blockchain still lags when compared with transaction-based technologies such as Visa, which is capable of handling thousands of transactions per second on average. In contrast, a Bitcoin blockchain, is tremendously lower in both transaction speed and volume. Although several solutions are currently underway to address the limited scalability of blockchain, they are still in a nascent development stage. As an illustration, the use of off-chain data storage can improve the efficiency of blockchain; however, this solution may surface new issues related to data integrity and privacy [114]. Therefore, the application of blockchain in the FSC depends on high scalability and decentralization [103] so that FSC operations become smoother and more flexible.

Blockchain-based FSCs require more intelligence and automation to increase the resiliency of the food supply network by lessening the restrictions for users to join, execute transactions, and access smart contracts' source codes [115]. Code errors and security vulnerabilities might result in significant financial losses for FSC actors. Wu et al. [114] claim that blockchain might be subject to several security threats such as a mining attack, which can put food companies at risk of data and revenue loss, whereas Zhao et al. [100] highlight that the decentralized architecture of blockchain and its integration with an extensive peer-to-peer wireless sensor network might give rise to several privacy and security issues in the agri-food value chain. Zhao et al. [100] claim that operating a blockchain may entail an unprecedented level of transparency and visibility of FSC processes leading to a potential risk in confidentiality. Blockchains may not guarantee privacy for FSC actors as information will be available and accessible to all members belonging to a network. If information is deemed strategic, confidential or secret, food firms might be hesitant to engage in blockchain-based FSCs until the risk can be mitigated [114].

### 4.2.2. Organizational Challenges

The technological immaturity of blockchains is a significant barrier to its adoption in the FSC alongside the nascent nature of research and development [121]. The immaturity level of the technology exacerbates the uncertainties concerning its usefulness for FSC activities [127]. Many blockchain projects in the food industry are in a proof-of-concept stage of development. As such, many blockchain start-ups and established technology firms have introduced solutions based on blockchain technology, but not many applications have gone beyond a pilot stage [1]. Moreover, food organizations interested in adopting blockchain might be hesitant to place the technology at the core of their organizational processes due to the lack of previous experiences. Longo et al. [47] state that FSC players, such as food manufacturers, logistics operators and especially small- and medium-sized food enterprises, must invest more resources and money in harnessing and operating a blockchain-enabled supply chain. The initial set up and maintenance costs might outweigh the benefits of such FSCs. Firms with limited financial resources might be unable to increase their adaptation capabilities, thereby being at a competitive disadvantage compared to their competitors. Failure to understand that the adoption of blockchain needs enhanced organizational capabilities can result in severe operational problems.

In terms of implementation costs, the use of blockchain might require significant capital investments [158]. Klerkx et al. [127] argue that the digitalization of supply chain processes requires the mobilization of diverse skills, knowledge, and materials to translate digital data and capabilities into better decisions for FSC management. The novelty of blockchain technology presents additional challenges for FSC operators because they may be unfamiliar with the principles (e.g., immutability), functioning, and maintenance of the technology [100]. Therefore, the need arises to develop workers' skills and technical understanding and capabilities to ensure effective implementation. Moreover, in the digitalization of FSC processes and the automation of manual roles, blockchain technology may force worker layoffs and poses a threat to the digitally illiterate [127].

### 4.2.3. Regulatory Challenges

To ensure the sustainable functioning of blockchain-enabled FSCs, food industry stakeholders have to follow regional, national and international policies and regulations [85]. Government support and regulations can be a driving force for the greater diffusion of the technology. Furthermore, blockchain technology implementation should be seen as a facilitator for regulatory and certification norms, especially those requiring the traceability of food products [80]. The lack of industry standards related to blockchain technology makes it hard to integrate all FSC exchange partners into a single regulatory framework. However, the GS1 EPCIS-enabled protocol from Origin Trail offers an open-source solution to address the problem of blockchain to blockchain and blockchain to legacy interoperability (other solutions may also exist). Cooperation among different industry players needs to be fostered to ensure greater regulatory harmonization. GS1 standards are also intended to aid FSCs to collaborate on specific aspects of regulatory compliance (e.g., traceability, recall, product and trading party identity, farm or factory identity and location, labelling). Misconceptions regarding blockchain technology need to be addressed as they may result in regulatory and legal restrictions and limit the value obtained from adopting the technology [80]. In this respect, Behnke and Janssen [1] maintain that standardization of food traceability processes and the development of a unified framework is a crucial boundary requirement before blockchain can be applied to the FSC. As a result, blockchain promoters in governmental roles need to involve technical experts to establish the regulation upon which blockchain technology can be developed.

## 5. Discussion, Implications and Conclusions

### 5.1. General Discussion

In this paper, we synthesize the current knowledge base concerning the potentials and challenges of blockchain in the FSC. We report findings from an SLR and bibliometric analysis showcasing the

role of blockchain in the food industry. The network and content analysis results show that blockchain is a promising technology and an effective solution for modernizing FSCs [159]. The intrinsic characteristics of blockchain have the potential to solve several problems inherent to FSCs. For example, food companies using blockchain technology can enhance product traceability and quickly identify the location and source of products implicated in a food safety recall, adulteration, or counterfeit. Numerous studies advocate using blockchain as a means for achieving end-to-end product traceability, making the tracking and tracing of food products feasible at every point in the supply chain [160]. It can be noted that traceability of food products can be done without blockchain. However, it is often cumbersome and inefficient when multiple stakeholders are involved due to a frequent lack of interoperability and the limitations of a one-up/one-down regulatory requirement for traceability. Blockchain-based traceability can help determine the different actors' identity, food product identity, origins, and all related information [161]. From a supply chain perspective, blockchain will help to ensure efficient and trusted transactions while simultaneously enhancing traceability and recall capability, aiding food safety and the rapid identification of potential food fraud, counterfeits and other forms of illicit trade [162]. Blockchain can reduce information asymmetry and provide more accurate, timely and trusted information to the public while minimizes damage to a company's brand image and reputation.

Our findings also highlight the value of blockchain for FSC collaboration. More specifically, blockchain provides opportunities for FSC stakeholders to collaborate more efficiently and effectively, increasing trust in business interactions. In FSCs, trust is a precursor or antecedent for successful collaborative arrangements and vital for information sharing and transparency [163]. As a result, food companies can rely on blockchain to forge closer partner collaboration, sustain FSC activities, and mitigate the adverse effects of process failures. Similarly, blockchain deployment for FSC collaboration can be an impetus for higher supply chain responsiveness and organizational performance. Our findings suggest that blockchain may lead to better performance in process automation, information sharing and decision synchronization. FSC stakeholders can benefit from using blockchain to foster collaboration among the different tiers of the food chain since the exchange of information and resources is independently verified by blockchain participants and can be inspected by food suppliers, distributors and customers. It is also conceivable that the technology can be a means by which all FSC stakeholders can be involved in a collaborative environment that promotes shared responsibility, fairness and transparency [163].

Blockchain is anticipated to synchronize information sharing among FSC stakeholders to significantly reduce excess inventory and protect against the harmful bullwhip effect. Thanks to the increased visibility of FSC processes, blockchain helps the food industry stakeholders overcome the common problems of conventional collaborative systems by enhancing response time, increasing cost-effectiveness, reducing potential errors, and assuring instant availability of accurate and reliable FSC information [164]. Therefore, it is essential to integrate blockchain in FSC collaboration to establish trust in the relationships among FSC partners, which is a critical element for achieving successful collaborative practices. Successful collaborative practices also open opportunities for social sustainability. When it comes to human rights and fair work, a complete record of a product's history helps product buyers be confident that their purchase originates from ethically sound sources.

Several studies stress blockchain's ability to improve the operational efficiencies of FSCs. In this regard, the most apparent feature of blockchain to increase efficiency is disintermediation by helping FSC partners automate business transactions, reduce lead times, and reduce costs. Blockchain can create an ecosystem wherein frictionless value transfers can be performed efficiently. The automation of FSC transactions paves the way to optimizing FSC processes such as food sourcing, ordering and distribution, thereby helping businesses identify potential sources of process inefficiencies, redundant tasks and fraud. The virtues of getting an improved view of the flow of food products can aid FSC partners in managing their production, inventory, and food safety mechanisms, thereby reducing food waste and spoilage. Blockchain coordinates FSC operations to enable faster customer responsiveness and

maximize customer service levels. It may help reduce the size and scope of rework and recalls by delivering these services, thus providing considerable greenhouse gas reductions and other resource savings. Access to more complete longitudinal supply chain datasets will lead to improved practices, including eliminating redundancies and bottlenecks, and ultimately, decreases in resource consumption, all of which are positive outcomes of blockchain technology from a sustainability perspective [165].

Additionally, FSC stakeholders must consider the integration of blockchain in their trading processes. The opportunity to develop a fair-trade environment is frequently highlighted in the reviewed literature. The persistent inefficiencies associated with the movement of food products from one country to another can be partially resolved with blockchain. In this sense, blockchain can simplify food trade logistics and distribution, eliminating information asymmetry and establishing a more credible and sustainable food trading environment.

Our review of the literature also scrutinizes the challenges encountered when deploying technology in the food industry. For example, the technical limitations of blockchain, including scalability, security, and privacy issues [166], represent a significant hindrance to its application in FSCs. In practice, scalability is crucial when deciding to deploy blockchain because the technology might not be suitable for managing FSC data, especially when substantial, high-velocity information needs to be processed. The storage capacity and performance of blockchain might not work for data-intensive FSCs since all blocks must store a copy of all transaction data fed into the network, resulting in data redundancy. Although several solutions have been introduced to improve scalability, further efforts are needed to develop highly scalable blockchains that respond to the needs of all FSC stakeholders. Furthermore, using blockchain in a multi-tier FSC network poses additional risks for security attacks and privacy intrusions [167,168] due to the technology's decentralized architecture.

Consequently, if FSC partners feel that their business information is not secure, they will be discouraged from using blockchain. Previous research also shows that the adoption of blockchain in the FSC may be slowed down by many organizational factors such as technology immaturity, resistance to change and the lack of necessary resources and operational capabilities [169]. Applying blockchain for FSCs is still a challenging endeavor as the technology and its design are still unable to cope with highly globalized FSCs. The uncertainties surrounding the technology and the fear of losing control may explain many managers' reluctance to support blockchain-enabled business models. FSC actors might be unwilling to operate in a blockchain environment [170] if their competitors develop a competitive edge by concealing a particular product or processing information. Moreover, the commitment of significant resources for the engagement in a blockchain operational model is highlighted in the literature as a pressing challenge for small and budget-constrained FSC partners because they are required to incur additional costs for organizational development capabilities and system maintenance. Lastly, the literature also discusses regulatory issues facing blockchain adoption in FSCs [55]. The legal environment of blockchain is still full of uncertainties [171]. For example, there is a need for regulations and industry standards that FSC stakeholders can refer to when encountering potential incidents while operating in a blockchain setting. Therefore, industry standards and regulatory initiatives are necessary to accelerate blockchain adoption in FSCs and develop best practices and protocols for FSC interoperability.

### 5.2. Implications for Researchers and Practitioners

The pressing need for maintaining highly efficient, integrated and responsive FSCs is driving researchers and organizations to rethink the supply chain design. Substantial efforts are devoted to studying promising opportunities for this technology in the food industry. The findings of this study are useful for researchers to capture the dynamics surrounding blockchain technology. More specifically, we unfold the potential areas where food organizations can use blockchain to add substantial business value and achieve sustainable performance. Reviewing the potentials and challenges of blockchain is crucial for leading FSC stakeholders, who need to scrutinize the technology enablers to create strategies and policies, incentivizing the transition from conventional FSC systems to blockchain.

Innovation in blockchain through improved scalability, performance and security, can contribute to the wide-scale implementation of the technology in FSCs. Regulatory bodies and key FSC stakeholders that exert pressure on transitioning toward blockchain-driven FSCs need to develop policies and standards that facilitate regulatory document expedition and streamline business processes, such as the checking, control, control, monitoring and certification of food products. The shift toward a blockchain ecosystem may necessitate a different modus operandi regarding the orchestration of FSCs. Therefore, we recommend that food organizations and stakeholders diminish the main barriers to blockchain adoption at the organizational level (i.e., resistance to change) and develop the organization's needed operational capabilities by providing specialized corporate training programs and developing systems to enhance coordination and information sharing between FSC partners. Blockchain has the potential to empower the technological advancement of FSCs as it can augment the capabilities of other Industry 4.0 technologies, such as IoT, Big Data, Artificial Intelligence, Augmented Reality and Cyber-Physical Systems to generate new efficiencies. Thus, managers and practitioners need to be cognizant of the transition toward blockchain-enabled FSCs to deliver high quality, safe and authentic food to consumers.

Managers should also be aware of the additional challenges and tensions that can emerge from blockchain adoption, and they should work to overcome issues threatening the sustainable functioning of FSCs. This study reveals that blockchain adoption in FSCs is worthwhile in terms of food traceability, collaborative relationships, operational efficiencies, and food trade activities. Thus, managers need to ensure their digital transformation plans consider the potential transformative power of blockchain-based business models. Investments in blockchain promise improved food traceability, trust, transparency and efficient use of resources, and strong relationships with customers and supply chain partners, which can help food firms sustain their competitive advantage.

Our literature review framework provides a comprehensive analysis of opportunities and constraints of blockchain in the FSC that drives or impedes sustainable management to inspire further research. We propose the following research gaps that need further attention and investigation:

- This review highlights the potentials of blockchain for FSCs. However, insufficient attention has been paid to the role of blockchain in supporting internal activities within food organizations, namely, raw materials procurement, inventory management, document and credentials management, specification and recipe management, product life cycle management, quality management and the role of smart contracts.

- Additional studies on the role of blockchain in FSC collaboration are required to understand better the tensions and paradoxes that can arise from the technology's integration and interoperability in complex and multi-tier FSCs.

- Empirical studies are required to test whether the technological capabilities of blockchain can enable and constrain FSC performance.

- Our findings illustrate that blockchain helps to improve FSC processes. However, exactly how blockchain can help to overcome problems and bottlenecks of organizational performance remains unknown. Another important research topic is examining the impact of blockchain on FSC resource sharing, decision synchronization and joint knowledge creation.

- Future research needs to provide a quantitative assessment of the impact of blockchain on FSC performance and provide clear guidelines on how to tailor blockchain characteristics to increase the efficiency of FSCs and respond to the needs of all stakeholders involved in the food industry. The framework that emerged from the literature analysis can be a starting tool to map the different needs of FSC partners and introduce appropriate blockchain solutions to respond to concerns in terms of food security, safety and convenience using technology.

- Future research needs to discern workable solutions to overcome the technical, organizational, and regulatory challenges facing blockchain implementation in FSCs.

- Future studies need to investigate the impact of blockchain on consumer purchasing habits and consumption of food products. Additionally, researchers should focus on the use of blockchain to design mechanisms for more sustainable and ethical food production, thereby improving consumer satisfaction and trust in food products.
- Future studies need to examine blockchain's added value when used together with forensic testing methods to ensure food authenticity, provenance, and safety.
- Additional case studies need to be conducted to validate the diverse themes of our framework and highlight the applicability and suitability of blockchain to diverse areas in the FSC.
- Researchers are required to elaborate on blockchain's role to foster FSC sustainability, detailing the impact of the technology on economic, social and environmental dimensions of FSC sustainability. Addressing this knowledge gap is necessary to grasp the transformational impact of technology on the economy and society.

### 5.3. Conclusions

An essential task of a literature review is to provide a timely synthesis and analysis of published literature. Even though previous studies have explored the numerous possibilities of blockchain technology in supply chain management, they have not captured the latest technology developments from the FSC perspective. Therefore, our review study aims to enhance scholars' understanding regarding the potentials and challenges of leveraging blockchain in the food industry.

In this study, we employ an SLR and investigate the current state of knowledge on blockchain applications in the FSC. The review was conducted with sixty-one (61) relevant journal articles, which were thoroughly examined and analyzed using bibliometric tools and techniques. This study reveals that blockchain technology is still in a nascent stage and has a potentially transformational and foundational (rather than disruptive) impact on the FSC. As for the benefits of the technology, we found that blockchain adoption can improve food traceability, enhance FSC collaborative relationships, maximize operational efficiencies, and sustain food trading activities. The downsides of blockchain mainly fall under three categories, namely, technical, organizational and regulatory barriers. Issues, including blockchain scalability, security and privacy, are the key factors inhibiting the widespread implementation of the technology. The lack of standards and regulatory support is also expected to restrain blockchain's value in the food industry.

Through conducting this review, our primary goal is to inform scholars and practitioners on the importance of blockchain technology in sustaining FSCs. Moreover, we seek to summarize the current research state and provide several implications for researchers and practitioners. Furthermore, the compilation of our review and its findings should encourage further research in the field. Aside from offering valuable contributions to the blockchain literature and deepening the extant literature's overall understanding, we highlighted the main knowledge domains.

From a theoretical perspective, we provide three contributions. First, this paper adds to the few studies that have previously explored blockchain technology in the FSC. Second, we synthesize related literature using keyword co-occurrence and bibliographic coupling techniques. So far, bibliometric methods have not fully exploited the review of blockchain research in the food area. Hence, this study offers a detailed analysis and timely synthesis of the literature. Third, our findings identify several areas that are not sufficiently dealt with, such as blockchain technology's role in enhancing FSC sustainability through better collaborations with partners in multi-tier global food supply networks.

This review has several practical implications. For instance, blockchain technology benefits may provide a reference for practitioners interested in understanding the expected outcomes from the deployment of the technology in the FSC. The welfare of various FSC stakeholders such as food suppliers, producers, retailers and final consumers may be substantially improved with blockchain technology. In this regard, this review can help practitioners to understand these far-reaching implications better. In contrast, blockchain challenges may guide FSC managers to identify the pain points encountered by organizations during the shift toward blockchain-enabled FSCs.

As with any research, this study has some limitations. The selection of Scopus for the collection of the literature does not guarantee the full coverage of research works published on blockchain technology from the FSC context. The final list of retrieved articles was generated based on the set of search keywords used. Although keyword entries provide a comprehensive list of research articles, the remaining keywords may not be exhaustive. Future studies may want to consider using other databases such as ISI Web of Science. We also recommend that researchers empirically validate the research questions raised in this review using surveys and case studies.

**Author Contributions:** Conceptualization, A.R. and K.R.; methodology, A.R., S.Z., H.T.; software, K.R.; validation, J.G.K., S.Z. and H.T.; formal analysis, A.R.; investigation, A.R.; resources, A.R., K.R.; data curation, K.R.; writing—original draft preparation, A.R.; writing—review and editing, A.R., J.G.K., S.Z., H.T.; visualization, K.R.; supervision, J.G.K., S.Z., H.T.; project administration, H.T. All authors have read and agreed to the published version of the manuscript.

**Funding:** This research received no external funding.

**Acknowledgments:** A.R. is grateful to László Imre Komlósi, Katalin Czakó and Tihana Vasic for their valuable support.

**Conflicts of Interest:** The authors declare no conflict of interest. Author Keogh acknowledges his role as a former advisor to not-for-profit organization 'Origin Trail' and completed the assignment in early 2019. Keogh held executive roles at not-for-profit standards body GS1, completing that work more than 5-years before this publication.

## Appendix A

TITLE-ABS-KEY (blockchain* AND (food* OR agriculture OR agri-
food OR farming OR "cold chain*" OR "fresh product*" OR "agri-
fresh" OR vegetable* OR fruit* OR perishable)) AND (LIMIT-
TO (DOCTYPE, "ar") OR LIMIT-TO (DOCTYPE, "re")) AND (LIMIT-
TO (SUBJAREA, "BUSI") OR LIMIT-TO (SUBJAREA, "COMP") OR LIMIT-
TO (SUBJAREA, "ENGI") OR LIMIT-TO (SUBJAREA, "DECI") OR LIMIT-
TO (SUBJAREA, "SOCI") OR LIMIT-TO (SUBJAREA, "AGRI") OR LIMIT-
TO (SUBJAREA, "ENVI") OR LIMIT-TO (SUBJAREA, "ECON") OR LIMIT-
TO (SUBJAREA, "MULT"))

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
