# Peer review of "Blockchain Technology in the Food Industry: A Review of Potentials, Challenges and Future Research Directions"

_logistics, 2020_

Round 1

Reviewer 1 Report

An important weakness of the article is the lack of empirical research results and references to practical solutions.

Reviewer 2 Report

The topic of the paper is actual and interesting.

Semantic analysis of the literature in section 3.4.2. Knowledge Domains through Bibliographic Coupling is a little bit confusing. Please try to add despite of numbers for particular papers also your author's comparison regarding key words, citation index. In this form such analysis is difficult to understand what was the main objective of it

Figure 6 could be developed maybe in the form of mind map, such graphics is too trivial. 

The number of references presented in sections: 4.1.1. - 4.1.4. should be developed. 

I recommend report: https://www.accenture.com/_acnmedia/PDF-93/Accenture-Tracing-Supply-Chain-Blockchain-Study-PoV.pdf

Alfaro, J. A., & Rábade, L. A. (2009). Traceability as a strategic tool to improve inventory management: a case study in the food industry. International Journal of Production Economics, 118(1), 104-110.

Aung, M. M., & Chang, Y. S. (2014). Traceability in a food supply chain: Safety and quality perspectives. Food control, 39, 172-184.

Charlebois, S., Sterling, B., Haratifar, S., & Naing, S. K. (2014). Comparison of global food traceability regulations and requirements. Comprehensive reviews in food science and food safety, 13(5), 1104-1123.

Gartner (2019a). Gartner Predicts 20% of Top Global Grocers Will Use Blockchain for Food Safety and Traceability by 2025. Available Online:
https://www.gartner.com/en/newsroom/press-releases/2019-04-30-gartner-predicts-20-percent-of-top-global-grocers-wil 

Despite literature reviewing it would be valuable to describe how blockchain technology is evolving with respect to food traceability and give to readers some  detailed information about technology etc.

Especially since title of the paper suggest that authors provide more information than solely comparison of literature in blockchain and food industry. 

Reviewer 3 Report

The selection of Scopus for the collection of the literature does not guarantee the full coverage of research works published on block chain technology from the FSC context. You should add some research using other international databases: Clarivate - ISI Web of Science, DOAJ, etc

What is the impact of your research for Researchers and Practitioners? Please improve.

The paper is a nothing else but a literature review. The contribution of the paper is low.

How can the conceptual model - Fig 6 be validated?
(Figure 6. A conceptual framework for the literature analysis.)

Conclusions can be improved - please highlight better the future work - scientific work!.

Round 2

Reviewer 1 Report

    The text may be published after the final linguistic revision.

Author Response

Dear Reviewer,

Thank you very much for your feedback. Our paper was carefully proofread by a native speaker and we hope that you are satisfied with the final version.

-- The Authors

Reviewer 3 Report

A case study in an organization based on your framework - fig 6. can be done - as a recommendation.

Author Response

Dear Reviewer,

Thank you very much for your feedback. We included your suggestion regarding the case study as a suggestion for future research.

Best regards,

-- The Authors